# Structural basis for DARC binding in reticulocyte invasion by *Plasmodium vivax*

Re'em Moskovitz[1,3], Tossapol Pholcharee[1,3], Sophia M. DonVito [2], Bora Guloglu [1], Edward Lowe [1], Franziska Mohring[2], Robert W. Moon [2] & Matthew K. Higgins [1] ✉

The symptoms of malaria occur during the blood stage of infection, when the parasite replicates within human red blood cells. The human malaria parasite, *Plasmodium vivax*, selectively invades reticulocytes in a process which requires an interaction between the ectodomain of the human DARC receptor and the *Plasmodium vivax* Duffy-binding protein, PvDBP. Previous studies have revealed that a small helical peptide from DARC binds to region II of PvDBP (PvDBP-RII). However, it is also known that sulphation of tyrosine residues on DARC affects its binding to PvDBP and these residues were not observed in previous structures. We therefore present the structure of PvDBP-RII bound to sulphated DARC peptide, showing that a sulphate on tyrosine 41 binds to a charged pocket on PvDBP-RII. We use molecular dynamics simulations, affinity measurements and growth-inhibition experiments in parasites to confirm the importance of this interaction. We also reveal the epitope for vaccine-elicited growth-inhibitory antibody DB1. This provides a complete understanding of the binding of PvDBP-RII to DARC and will guide the design of vaccines and therapeutics to target this essential interaction.

In many regions of the world, *Plasmodium vivax* is the predominant parasite that causes human malaria. Leading to around 14.5 million diagnosed cases each year, it accounted for an estimated 61% of malaria in the Americas and 48% in South East Asia in 2017[1]. With studies suggesting that the impact of this parasite is underestimated[2,3], morbidity caused by *P. vivax* is a major global health problem and an effective vaccine is urgently required to contribute to the malaria eradication effort.

The symptoms of malaria occur when parasites replicate within the blood. In the case of *P. vivax*, invasion of red blood cells is highly dependent on the reticulocyte Duffy antigen/receptor for chemokines (DARC). The importance of DARC as an invasion receptor[4] was first shown over 45 years ago, with the demonstration that the related parasite, *Plasmodium knowlesi*, is unable to invade blood cells taken from individuals with the Duffy-negative phenotype[5]. These people have a mutation in their DARC gene which makes the receptor undetectable on circulating reticulocytes. Duffy negativity was also found to

prevent African-Americans from being infected with *P. vivax* through mosquito bite[6]. The widespread prevalence of *P. vivax* across the world, together with its much-reduced occurrence in regions of Africa where Duffy-negativity is widespread, highlight the importance of DARC as a determinant of *P. vivax* susceptibility[7]. Indeed, while *P. vivax* is occasionally capable of infecting Duffy-negative individuals, this is linked to lower parasite load and more than 15-fold reduced likelihood of causing clinical disease[8–10]. Preventing DARC-mediated invasion remains a primary approach to prevent clinical vivax malaria.

The parasite binding partner for DARC is the Duffy binding protein, PvDBP[11] and multiple lines of evidence highlight the important nature of their interaction. In the closely related, genetically tractable parasite, *P. knowlesi*, knockout of PkDBPα prevents invasion of Duffy-positive erythrocytes in vitro[12–14]. While PvDBP knockout in *P. vivax* has not been technically possible, due to challenges with maintaining this parasite in culture conditions, immunisation of mice, rabbits and non-human primates with the RII region of PvDBP (PvDBP-RII) induces

[1]Department of Biochemistry, University of Oxford, South Parks Road, Oxford OX1 3QU, UK. [2]London School of Hygiene and Tropical Medicine, Keppel Street, London WC1E 7HT, UK. [3]These authors contributed equally: Re'em Moskovitz, Tossapol Pholcharee. ✉e-mail: matthew.higgins@bioch.ox.ac.uk

inhibitory antibodies that block binding of PvDBP to DARC[15,16]. In humans, high titres of naturally acquired antibodies that target PvDBP-RII and prevent DARC binding in vitro, are associated with reduced risk of *P. vivax* infection[17], lower parasite densities following invasion and decreased risk of clinical malaria[18,19]. Immunisation of human volunteers with recombinant viral vectors expressing PvDBP-RII induces strain-transcending antibodies which prevent recombinant PvDBP-RII from binding to DARC while human antibodies, derived from either vaccination or natural infection, are inhibitory of invasion[13,20]. PvDBP is therefore the primary blood stage vaccine candidate to prevent vivax malaria.

With both intact PvDBP and DARC proving challenging to produce, molecular studies have used smaller fragments to narrow down functionally important regions[21], with a ~ 350 amino acid residue Duffy-binding-like (DBL) domain known as PvDBP-RII, shown to bind DARC[22]. PvDBP-RII binds to the sixty-residue N-terminal ectodomain of DARC[23]. The first structural study related to PvDBP was a crystal structure of PkDBP-RII, from *P. knowlesi*[24]. This was followed by structures of *Plamodium vivax* DBP-RII bound to a 30 residue-long peptide (DARC$_{14-43}$), of which 11 residues (DARC$_{19-30}$) were ordered and were observed in the structure as a helical peptide[25].

However, DARC is post-translationally modified by sulfation of two tyrosine residues, Y30 and Y41 and mutation of Y41 to phenylalanine reduced PvDBP binding[26]. These residues are not observed in existing structures, most likely because the DARC ectodomain used was bacterially expressed and therefore lacked tyrosine sulfation[26,27]. Studies of *P. knowlesi* DBP-RII also proposed the locations for binding sites of the sulfated tyrosines, with positively charged patches on PkDBP-RII suggested to form the binding sites for Y41[24] and Y30[28]. Indeed, a later refinement of the PkDBP-RII structure showed a sulfate ion bound in the proposed binding site for Y41[28]. Mutations have also been described in regions of PvDBP which do not contact DARC$_{19-30}$ but which are known to contribute to full-length DARC binding[29,30] and a polymorphism in residue 42 of DARC is the sole change in the Fy$^a$ blood group, which reduces the likelihood and severity of *P. vivax* infection[31]. Indeed, NMR experiments indicate that residues from 14-43 of DARC show chemical shifts on binding to PvDBP-RII[25]. The full interaction between PvDBP and DARC therefore remains to be defined.

Structural approaches have also started to reveal how the PvDBP-DARC interaction can be blocked by mouse and human monoclonal antibodies. Two human monoclonals bind the region of PvDBP involved in DARC$_{19-30}$ binding and dimerisation[20]. In contrast, a broadly inhibitory human monoclonal, from a vaccinated human volunteer, binds away from this site[13], as do inhibitory mouse antibodies[32]. The molecular mechanisms by which these antibodies work are uncertain.

In this study, we determine the structural basis for the interaction of PvDBP with the DARC ectodomain and the human invasion blocking antibody, DB1. By using DARC proteins expressed in human cells, we define the binding site for sulfated tyrosine 41. This reveals the molecular basis for the role of tyrosine sulfation in the essential interaction between PvDBP and DARC in reticulocyte invasion by *P. vivax*.

## Results
### Structural characterisation of the PvDBP-DARC interaction
To determine the structure of PvDBP-RII bound to sulfated DARC ectodomain we started by expressing the N-terminal 60 residues of DARC (DARC$_{ecto}$) in HEK293 cells. This was purified and assessed by mass spectrometry, with the predominant peak having the mass expected for the peptide with the addition of two sulfates (Supplementary Fig. 1). We also expressed PvDBP-RII in HEK293 cells and this was mixed with DARC$_{ecto}$ for crystallisation. As no crystals formed, we attempted to use Fab fragments from monoclonal antibodies which bind to PvDBP-RII, but do not prevent it from binding to DARC$_{ecto}$, as crystallisation chaperones[13]. A complex of PvDBP-RII, DARC$_{ecto}$ and the

Fab fragment of the DB1 monoclonal antibody[13] formed crystals which diffracted to 2.49 Å resolution. Molecular replacement using the PvDBP-RII[13] structure as a search model provided phase information and the resultant electron density map revealed density for residues 19-47 of DARC$_{ecto}$ (Fig. 1 and Supplementary Table 1).

A comparison with the previously determined structure[25] of PvDBP-RII bound to DARC$_{ecto}$ expressed in bacteria shows DARC residues 19-30 to adopt the same binding mode in both cases, forming an α-helix which packs against subdomain 2 of PvDBP-RII, with a root mean square deviation of 0.22 Å over the helical residues 22-29 of DARC (Fig. 1b). While this is the only region of DARC observed in the previous study, we now also observe clear density for residues 31-47. These form an elongated peptide which wraps, in a horse-shoe-shaped trajectory, around a protrusion on the surface of PvDBP-RII subdomain 2. Clear density was observed for each residue, including a binding pocket for Y41 and the sulfate on Y41 (Fig. 1c), which closely matched a binding site previously proposed[28]. Also visible is the side chain of residue D42, which is polymorphic (D42G) in the Fy$^a$/Fy$^b$ blood group variant (Fig. 1c), but which does not make clear interactions with PvDBP-RII. Residues 48-60 are not visible in the crystal structure, suggesting a disordered linker between the PvDBP-RII-binding region of the ectodomain and the DARC transmembrane region, which starts at residue 61.

### Sulfated Y41 binds to a stable binding pocket on PvDBP while Y30 is dynamic
The structure allows us to assess the role of sulfation of Y30 and Y41 of DARC on the interaction with PvDBP. A previous study has shown that mutation of Y30 to F, which removes the hydroxyl group, as well as the potential for sulfation, did not affect PvDBP-RII binding[26]. Indeed, in our structure, the Y30 side chain was clearly visible in the electron density, but no evidence was seen for sulfation, suggesting either that the sulfate was not present, or that it was disordered in the electron density (Fig. 1d). The side chain did not contact PvDBP-RII, but instead interacts with a neighbouring DB1 Fab fragment through crystal contacts.

In contrast, the side chain of Y41, together with clear density for a sulfate group, was observed. This docks into a positively charged pocket on the PvDBP-RII surface, where the sulfate makes direct salt-bridges with K301 and R304 (Fig. 1c). This is consistent with sulfated Y41 (Y41-S) forming an important part of the binding interface. Indeed, the more substantial Y41F mutation, which removes both sulfate and hydroxyl group, reduces PvDBP binding[26] and this interaction is likely to be important for residues 31-47 to adopt their correct binding conformation.

We next assessed the stability of the binding modes of Y30 and Y41 using metadynamics simulations allowing us to generate the relative free energy landscapes of the side chains of sulfated and non-sulfated versions of Y30 and Y41[33,34]. These simulations were run for three different molecular models, generated from our crystal structure. These were sulfated on both Y30 and Y41, sulfated on just Y30 and sulfated on just Y41. In each case, we analysed the ensemble of structures generated during the simulation, the free energy surfaces and the number and duration of contacts formed between each tyrosine residue and PvDBP-RII (Fig. 2).

In the case of Y41, sulfation leads to an average free energy decrease of 7.63 kJ/mol with the decrease in free energy concentrated in the existing global minimum, at approximately $\chi_1 = -161°$, where we observed a local decrease of 17.87 kJ/mol, compared with $\chi_1 = -144.3°$ in the crystal structure (Fig. 2a). This corresponds with Y41 adopting a more preferred location when sulfated (Fig. 2b). In contrast, while sulfation of Y30 leads to an average free energy decrease of 8.75 kJ/mol, this changes the shape of the free energy surface, with the free energy barrier at approximately $\chi_1 = -120°$ disappearing (Fig. 2c) and Y30 being able to explore a

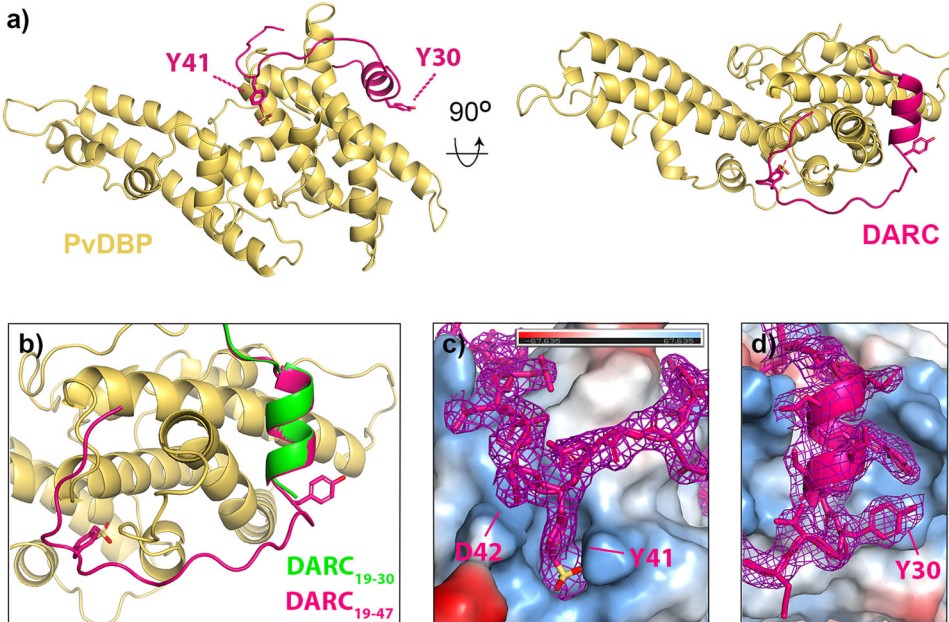

**Fig. 1 | The structure of PvDBP-RII bound to sulfated DARC ectodomain. a** The structure of PvDBP-RII (yellow) bound to the DARC ectodomain (pink). Residues Y30 and Y41 from DARC are highlighted as sticks, with sulfur in yellow and oxygen in red. **b** A close up of the DARC ectodomain, showing residues 19-47 of the sulfated ectodomain coloured as **a**), overlayed with residues 19-30 of the non-sulfated ectodomain in green (from PDB: 4NUV)[25]. **c** A close up of residue 41 of DARC, with DARC and the electron density surrounding DARC in pink and PvDBP-RII as a surface coloured by electrostatics (blue as positive charge and red as negative, estimated in pymol). **d** A close up of residue 30 of DARC, with DARC and the electron density surrounding DARC in pink and PvDBP-RII as a surface coloured by electrostatics (blue as positive charge and red as negative, estimated in pymol). In both **c**, **d**, the region of the $2F_O \cdot F_C$ map within 2 Å of DARC is shown at a contour level of 1.1.

wider set of $\chi_1$ configurations when sulfated (Fig. 2d). The location of Y30 in the crystal structure has $\chi_1 = 71.6°$, which lies close to the local energy minima at $\chi_1 = \sim 65°$ in the free energy landscape. Therefore, while the sulfate on Y30 is not observed in the crystal structure, and Y30 does not contact PvDBP-RII, simulations suggest that a flexible Y30S can interact with PvDBP-RII through a dynamic electrostatic interaction.

We next assessed the number of contacts which each residue from DARC forms with PvDBP and determined the fraction of the simulation during which each contact occurs. Using a heavy atom distance cut-off of 4.5 Å to define contacts, we observe that both Y30 and Y41 make more contacts with PvDBP when sulfated. This is much more pronounced for Y41 where the average number of contacts increases from 3.64 to 7.46, whereas in Y30 we observe a shift from 1.30 to 1.87 (Fig. 2e). We observe a similar pattern when considering the durability of specific interactions, with Y41 making more interactions with PvDBP than Y30 and with sulfation increasing both the number and durability of interactions (Fig. 2f). In particular, Y41 interacts continuously with R304 during the simulation, which it can only reach when sulfated. We therefore find that sulfation of Y41 stabilises a specific location for the side chain in which it makes strong electrostatic interactions, while sulfation of Y30 does not lead to the formation of such a favourable contact, but instead increases the strength and durability of a smaller number of transient interactions.

## Y41-S contributes more to binding affinity and erythrocyte invasion than Y30-S

The nature of the compact binding pocket for the sulfate on Y41, and the lack of a similar binding pocket for that on Y30, lead to the prediction that the sulfate on Y41 contributes more to the binding affinity than that on Y30. To test this, we generated a set of peptides containing the ordered region of DARC ectodomain (residues 19-47) and analysed their binding to PvDBP-RII by isothermal titration calorimetry (ITC) (Fig. 3a and Supplementary Fig. 2). A peptide in which both Y30 and Y41 are sulfated bound to PvDBP-RII with an affinity of 58 nM, while a peptide in which neither sulfate is present, bound with much lower affinity of 23 µM, quantifying the importance of sulfation for this interaction. We next tested peptides in which either Y30 or Y41 is sulfated. A peptide in which Y30 is sulfated, but Y41 is not, bound with an affinity of 552 nM, showing that the loss of Y41 sulfation leads to a ~9-fold reduction in affinity. In contrast, a peptide in which Y41 is sulfated but Y30 is not, bound with an affinity of 168 nM, with a ~3.5-fold reduction in binding affinity from the fully sulfated peptide. Therefore, both sulfates impact binding affinity, with Y41 providing a greater contribution to affinity.

We next assessed whether the differential effects of Y30 and Y41 sulfation on the affinity of DARC for PvDBP are mirrored during red blood cell invasion in vitro. To test this, we used a transgenic *P. knowlesi* line which has been modified to express PvDBP instead of the native, orthologous DARC binding protein, PkDBPα[13,14]. This transgenic parasite invades Duffy-positive erythrocytes in culture and is similarly affected by PvDBP-RII-targeting antibodies as *P. vivax* clinical isolates[13]. As it is currently technically impossible to generate transgenic erythrocyte lines in which DARC sulfation is specifically modulated, we instead assessed the ability of our four DARC ectodomain peptides to inhibit the growth of this transgenic PvDBP-expressing line, by measuring the effect of different concentrations of each peptide in a growth inhibition assay (Fig. 3b). We found that the Y30-S/Y41-S double sulfated peptide was most effective at blocking parasite growth ($IC_{50} = 0.72$ µM), followed by the Y41-S peptide ($IC_{50} = 2.99$ µM), with no growth inhibition observed for the Y30-S and non-sulfated peptides. Therefore, the effect of the peptides on parasite growth inhibition and the affinity that the peptides have for PvDBP-RII show the same pattern, with sulfation of Y41 having the greatest effect on peptide efficacy and sulfation of Y30 making a significant, but smaller difference.

We also conducted the equivalent experiment using unmodified (wild-type) *P. knowlesi* and observed a similar outcome, with

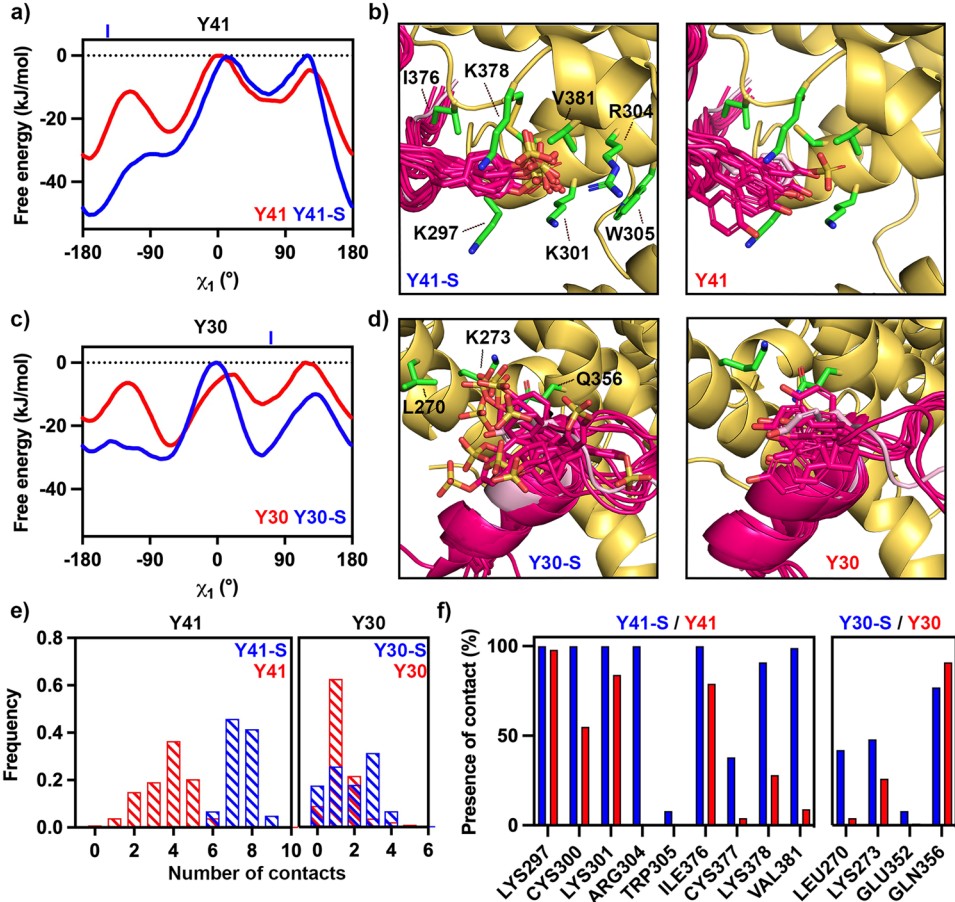

**Fig. 2 | Molecular dynamics simulations indicate ordered binding of Y41 but not Y30. a** Free energy landscapes for residue Y41 sulfated (blue) and non-sulfated (red) relative to the $\chi_1$ angle of the tyrosine, showing that sulfation favours a single binding position. In the crystal structure, $\chi_1 = -144.3°$, indicated by a blue line above the graph. **b** Representative images from across the simulation, showing the degree of motion of Y41 in its sulfated (left) and non-sulfated (right) forms. In each case, PvDBP is yellow and DARC is bright pink. The crystal structure is shown for comparison in light pink. In each case, sulfate atoms are yellow, oxygen atoms red and nitrogen atoms blue. **c** Free energy landscapes for residue Y30 sulfated (blue) and non-sulfated (red) relative to the $\chi_1$ angle of the tyrosine, showing that sulfation disfavours a single binding position. In the crystal structure, $\chi_1 = 71.6°$, indicated by a blue line above the graph. **d** Representative images from across the simulation, showing the degree of motion of Y30 in its sulfated (left) and non-sulfated (right) forms, coloured as in **b**). **e** The frequency of the number of contacts formed between PvDBP-RII and Y41 (left) and Y30 (right) in the sulfated (blue) and non-sulfated (red) forms during the simulations. **f** The percentage of frames from across the simulation in which each residue from PvDBP forms interactions with Y41 (left) and Y30 (right) in the sulfated (blue) and non-sulfated (red) forms. Source data are provided as a Source data file.

$IC_{50} = 0.79 \,\mu M$ for the double sulfated peptide, $IC_{50} = 3.92 \,\mu M$ for Y41-S and no inhibition observed for non-sulfated and Y30-S peptides, highlighting the parallel in DARC binding between PvDBP and PkDBPα. Therefore, while a panel of PvDBP RII-targeting antibodies which blocked invasion of *P. vivax* clinical isolates and PvDBP-expressing *P. knowlesi* (including DB1 and DB9) were not cross-reactive against wild-type *P. knowlesi*[13], the similar effect of DARC-based peptides on these two parasite lines indicates that PvDBP and PkDBPα engage DARC in a similar way.

**Mapping the epitope for growth-neutralising antibody DB1**

We next analysed the location of the epitope for DB1 (Fig. 4). This antibody inhibits growth of transgenic *P. knowlesi* parasites expressing PvDBP and of a subset of patient isolates of *P. vivax* which express different PvDBP variants[13]. Notably, DB1 binds to a site on PvDBP-RII which contains the polymorphic [339]DEK[341] motif[35], making direct contact with these residues (Supplementary Fig. 3), explaining in part why it is not a strain-transcendent neutralising antibody[13]. DB1 also binds to a site which does not overlap the DARC binding surface, explaining why it does not directly block DARC<sub>ecto</sub> binding[13] (Fig. 4a).

Previous structural studies of PvDBP-RII have observed dimers to form within crystals and have also shown DARC-dependent dimerisation at high concentrations in solution in the presence of a non-sulfated DARC peptide[25,27]. DB1 would block this proposed dimerisation interface, with the Fab fragment overlapping with the second PvDBP copy and associated DARC peptide from the dimer (Fig. 4b). To determine whether the mechanism of action of DB1 is inhibition of PvDBP-RII dimerisation, we used small-angle X-ray scattering to assess the solution mass of a complex consisting of PvDBP-RII bound to Fab fragment of DB1. As predicted, the PvDBP-RII:DB1 complex showed the mass and maximum dimensions expected for a monomeric complex (Supplementary Fig. 4). We also conducted SAXS experiment for PvDBP-RII bound to Y30-S Y41-S DARC<sub>19-47</sub>, but found that aggregation confounded collection of interpretable data. We therefore collected SAXS data for PvDBP-RII bound to Y30-S Y41-S DARC<sub>19-47</sub> and to the Fab fragment of antibody DB9. As DB9 binds distant from the dimerisation interface it would not prevent dimer formation. Both in SEC-SAXS experiments, in which the complex was passed through a size-exclusion column immediately before SAXS data collection (Supplementary Fig. 4), or when studied at high concentrations of up to 6 mg/ml in solution

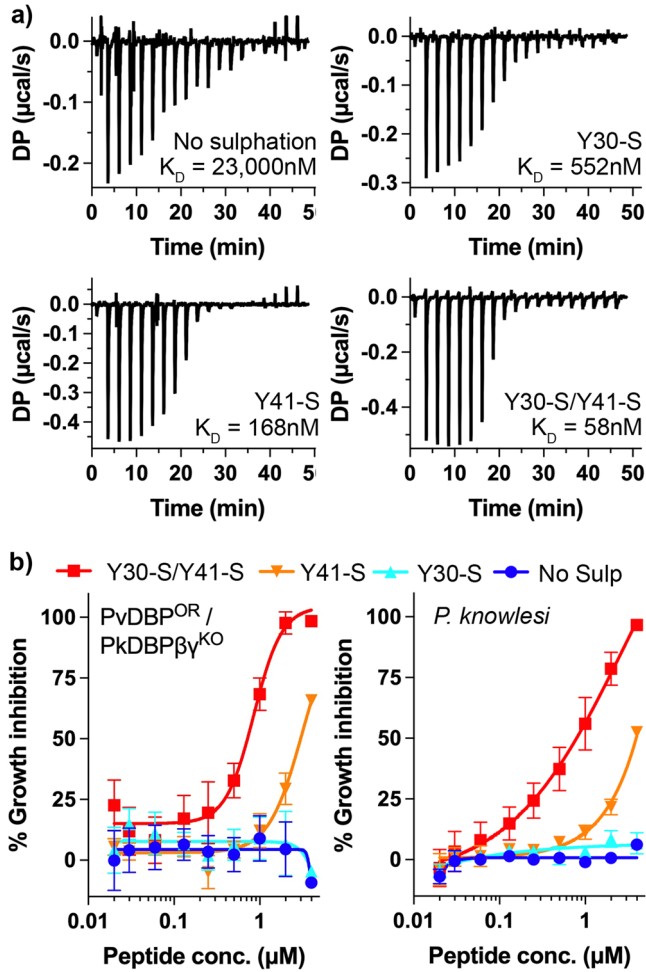

**Fig. 3 | The effect of tyrosine sulfation of DARC on PvDBP-RII affinity and parasite invasion. a** Isothermal titration calorimetry measurements of the binding of synthetic DARC peptides to PvDBP-RII. Shown are single representative traces for non-sulfated peptide and peptides sulfated on Y30, on Y41 or on both 30 and 41. Each stated $K_D$ value is the mean from n = 3 technical replicates. **b** Growth-inhibitory activity for the same four peptides in an assay which assess the growth of a *Plasmodium knowlesi* line in which PkDBPs have been replaced by PvDBP (PvDBP^OR/PkDBPβγ^KO, left) or wild-type *Plasmodium knowlesi* (right). The Y30-S/Y41-S peptide inhibited with an IC50 of 0.72 μM for PvDBP^OR/PkDBPβγ^KO and 0.79 μM for *P. knowlesi*. The Y41-S peptide inhibited with an IC50 of 2.99 μM for PvDBP^OR/PkDBPβγ^KO and 3.92 μM for *P. knowlesi*. Technical replicates (*n* = 2) from each assay were averaged, and data presented represents the mean ± standard error of the mean of four separate biological replicates (*n* = 2 in Fy^a donor blood, and *n* = 2 in Fy^b, to account for variation between DARC alleles). IC50 values were identified using a variable slope four-parameter logistic curve. Source data are provided as a Source data file.

(Supplementary Fig. 5) we did not observe dimerisation of the PvDBP-RII:DB9:DARC19-47 complex. As PvDBP-RII bound to sulfated DARC19-47 does not dimerise in our hands, we cannot therefore conclude that DB1 acts by blocking dimerisation.

## Discussion

Sulfation of DARC has been known for decades to play an important role in its binding to both cytokines and to PvDBP. However, previous structural studies used bacterially expressed DARC ectodomains, which lack tyrosine sulfation. To test whether these sulfates impact PvDBP binding, we therefore determined a structure of PvDBP-RII in complex with a sulfated DARC ectodomain. In the presence of tyrosine sulfation, we see an additional 17 residues of DARC, binding in an

extended conformation around a protrusion on the PvDBP surface. We found that the sulfate on Y41 docks into a compact pocket on PvDBP, making direct interactions which are likely to be required for residues 31-47 of DARC to become ordered. Indeed, specific removal of the sulfate on Y41 leads to a ~ 9-fold reduction in affinity and negates growth-inhibitory activity at the concentrations tested. In contrast, Y30 plays a less significant role in the interaction, not binding to PvDBP in the structure, dynamic in molecular dynamics simulations and with specific removal of the sulfate leading to ~3.5- and ~4-fold reductions in affinity and growth inhibitory activity.

Our study also reveals the epitope for the growth-neutralising antibody DB1 and shows that it binds to a polymorphic site distant from the DARC binding site, as predicted based on previous studies which show that DB1 does not block DARC binding[13]. Instead, DB1 binds in a location which would overlap with a putative dimerisation interface on PvDBP-RII[27]. The growth-neutralising antibodies which target PvDBP-RII, and have known structures, bind to a variety of locations, suggesting a range of possible molecular mechanisms for neutralisation (Fig. 4c). DB9[13] and 2D10[32] lie on subdomain 3, distant from both DARC binding site and the putative dimerisation interface, 092096[20] blocks both DARC binding and occludes the putative dimerisation interface and DB1 blocks only the putative dimerisation interface. However, our SAXS data revealed monomers of PvDBP-RII complexes in solution, even at high concentrations, leading us to question whether PvDBP-RII dimerisation is physiological and whether blocking of dimerisation is a mechanism of inhibition (Supplementary Figs. 4 and 5). Studies in which full-length PvDBP is assessed in the context of parasite invasion will be required to resolve whether dimerisation occurs in vivo.

Our study, therefore presents a complete view of the interaction between DARC and PvDBP-RII and reveals the role of DARC sulfation on PvDBP binding. However, our view of the complete PvDBP protein remains incomplete. Studies of PvDBP have long focused on the RII domain, through ease of production and due to the demonstration that this domain contains the DARC binding site. However, questions remain, which will only be answered when we understand the structure of full-length PvDBP and how it binds to full-length DARC. For example, it is also possible that the putative dimerisation interface observed in PvDBP-RII has other roles in full-length PvDBP, which are affected by antibody binding. Further study is also required to understand the role of the DARC Fy^a/Fy^b polymorphism on PvDBP binding as the role of the polymorphic residue 42 is not certain from this study. Nevertheless, we now reveal, for the first time, the full interaction between the ecto-domain of DARC and PvDBP-RII, showing the details of this interaction critical for red blood cell invasion in *P. vivax*. This identifies new surfaces of PvDBP which could be targeted by small molecules or antibodies that block the interaction with DARC, or that can be included in vaccine immunogens. These findings will guide future design of therapeutic agents to target the scourge of malaria.

## Methods

### Protein expression and purification

For crystallisation, the coding sequence for PvDBP-RII (D194-S508) was cloned into the pHLsec vector, which included a C-terminal 6xHis tag. DARC ectodomain (M1-S60, numbered as for the mature polypeptide) was cloned into the pENTR4 vector containing a C-terminal TEV protease cleavage site and 6xHis tag. Residues C4, C51 and C54 were changed to A to avoid disulfide formation and were not part of the PvDBP-RII binding site. DB1 heavy and light chains were supplied in the AbVec-hIgG1 and AbVec-hIgKappa vectors respectively. The constructs were transfected into Expi293 cells (Thermo Fisher Scientific) using the ExpiFectamin 293 Transfection Kit (Gibco). The transfected cells were collected after 4 days and the supernatant clarified by centrifugation at 10,000 × *g* and supplemented with 150 mM NaCl and 20 mM imidazole (final concentration).

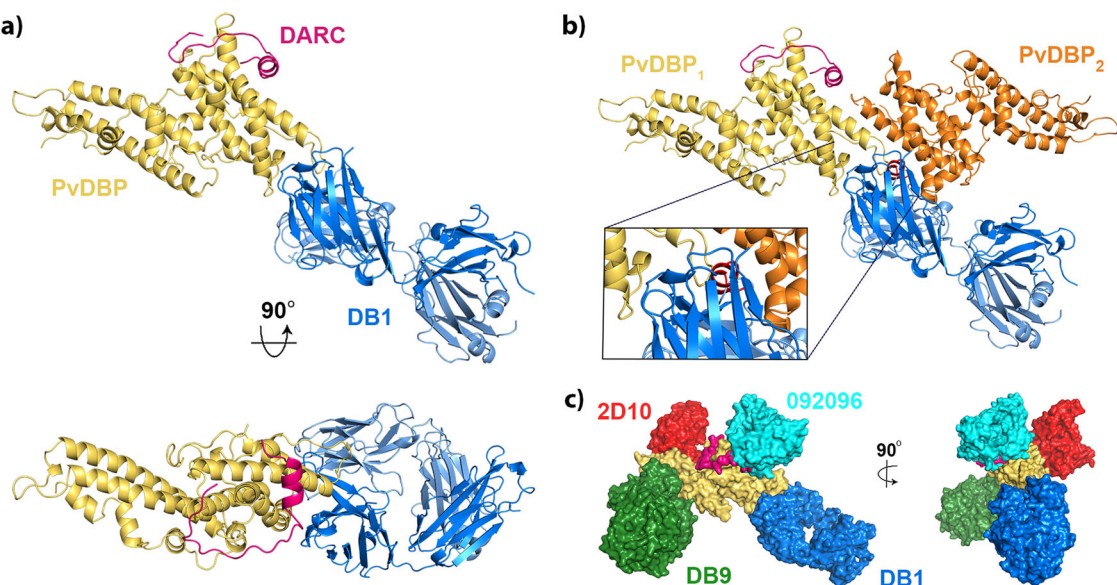

**Fig. 4 | Structural basis for neutralising antibody binding to PvDBP. a** Structure of PvDBP-RII (yellow) bound to DARC (pink) and antibody DB1. **b** A model of the putative PvDBP dimer (yellow and orange) bound to DARC and DB1, derived from PDB: 4NUV, showing that DB1 clashes with the putative dimerisation interface.

**c** A composite structure in which four different neutralising antibodies, DB1 (blue), DB9 (green)[13], 2D10 (red)[32] and 092096 (cyan)[20] are docked on to the structure of PvDBP-RII (yellow) and DARC (pink).

For isothermal titration calorimetry, PvDBP-RII (D194-T521) was expressed with an N-terminal monomeric Fc domain (mono-Fc). Three alanine substitution mutations, T257A, S353A and T422A, were introduced to the PvDBP-RII sequence to remove potential N-linked glycosylation sites. The construct was transfected into Expi293 cells as above and harvested after 5 days. The mono-Fc tagged PvDBP-RII was purified using HiTrap Protein A HP column (Cytiva).

DB1 antibody was purified by binding clarified cell supernatant to Protein A Agarose beads (Thermo Fisher Scientific), with elution into 100 mM glycine pH 3.0 and was quickly neutralised with 1 M Tris-HCl pH 8.0. To generate DB1 Fab fragments, purified DB1 was buffer exchanged into 150 mM NaCl, 20 mM HEPES pH 7.5, 20 mM L-cysteine for overnight cleavage at 37 °C with immobilised papain, followed by protein-A agarose purification to remove Fc and uncleaved DB1 antibody.

PvDBP-RII and DARC ectodomain used for crystallisation was purified by immobilised-metal-affinity chromatography by batch binding clarified cell supernatant to Ni-NTA resin, with elution into 300 mM imidazole, 150 mM NaCl, 20 mM HEPES pH 8.0 and was buffer exchanged into 150 mM NaCl, 20 mM HEPES pH 7.5 using a Vivaspin 3 kDa column. TEV protease and PNGase F were added to DARC ectodomain overnight at 4 °C to remove the TEV-6xHis tag and glycans, respectively.

**Crystallisation, data collection, and structure determination**

Crystallisation trials were carried out using vapour diffusion in sitting drops, mixing 100 nl protein complex solution and 100 nl well solution. Crystals were visible from day 12 with a well solution of 15% w/v jeffamine D-2003 and 10% v/v ethanol. The crystals were cryoprotected by transfer into drops of well solution supplemented with 25% glycerol and were cryo-cooled in liquid nitrogen for data collection.

Data was collected at the I24 beamline at Diamond at a wavelength of 0.9999 Å. After processing by autoPROC, the dataset was scaled using AIMLESS (v.0.73)[36], producing a complete dataset at a resolution of 2.5 Å. The structure was solved by molecular replacement with Phaser MR (v.2.8.3)[37] using a search model for PvDBP-RII (PDB: 6R2S). The model was built and refined using cycles of COOT (v.0.8.9.2)[38] and BUSTER (v.2.10)[39].

**Molecular dynamics simulations**

All simulations performed using OpenMM v7.7[40]. Models of PvDBP (residues 217-509) and DARC (residues 19-47) were capped at C- and N-termini using N-methyl and acetyl groups, respectively, protonated at pH of 7.5 using H++[41] and were soaked in truncated octahedral water boxes with a padding distance of 1 nm. NaCl was added to 150 mM while neutralising charges as described in[42]. Systems were parameterised using the Amber14-SB force field[43] and water modelled using TIP3P-FB[44] and tleap[45]. For sulfotyrosine residues, we used parameters from[46]. Three systems were prepared from our crystal structure, with both Y30 and Y41 sulfated, with only Y30 sulfated and with only Y41 sulfated.

Non-bonded interactions were calculated using the particle mesh Ewald method[47] with a real-space cut-off of 0.9 nm and error tolerance of 0.0005. Water molecules and heavy atom-hydrogen bonds were rigidified using SETTLE[48] and SHAKE[49] algorithms, respectively. Hydrogen mass repartitioning[50] was used to allow 4 fs time steps. Simulations were run using the mixed-precision CUDA platform in OpenMM using the Middle Langevin Integrator with a friction coefficient of 1 ps$^{-1}$ and the Monte-Carlo Barostat at a pressure of 1 atm. We equilibrated systems using a multi-step protocol: (i) energy minimisation over 10,000 steps, (ii) heating of the NVT ensembles from 100 K to 300 K over 200 ps, (iii) 200 ps simulation of the NPT ensembles at 300 K, (iii) cooling of the NVT ensembles from 300 K to 100 K over 200 ps, (iv) energy minimisation over 10,000 steps, (v) heating of the NVT ensembles from 100 K to 300 K over 200 ps, and (vi) 5 ns simulation of the NPT ensembles at 300 K.

We initialised well-tempered metadynamics using Plumed[51]. The first pair of simulations was biased on the $\chi_1$ and $\chi_2$ angles of DARC position 30. These were initialised with structures with both Y30 and Y41 sulfated and with just Y41 sulfated, using an initial hill height of 1.2 kJ/mol, a bias factor of 10, and bias widths of 0.07 rad, with biases being deposited every 500 steps. The second pair of simulations was biased on the $\chi_1$ and $\chi_2$ angles of DARC position 41. These were initialised with structures with both Y30 and Y41 sulfated and with just Y30 sulfated, using an initial hill height of 3.0 kJ/mol, a bias factor of 10, and bias widths of 0.07 rad, with biases being deposited every 500 steps. We found that a lower hill height of 1.2 kJ/mol was

insufficient to overcome free energy barriers in the system. For the second pair of simulations, we applied harmonic restraints on the backbone positions of the DARC peptide using the crystal structure as the reference and a force constant of 1500 kJ/(mol nm$^2$). Production simulations were performed for 500 ns and convergence was assessed using block error analysis on the computed free energy surfaces with respect to $\chi_1$ and $\chi_2$ angles of the biased positions. To recover unbiased ensembles from our simulations, we calculated weights ($w$) for each configuration ($s$) using the time-independent reweighting scheme (Eq. (1))[52]:

$$w(s) \propto \exp\left(\frac{V(s)}{k_{\mathrm{B}}T}\right) \qquad (1)$$

where $V(s)$ is the configuration-dependent bias at the end of the simulation and $k_{\mathrm{B}}T$ is the product of the Boltzmann constant and temperature. We used MDTraj (v. 1.9.6)[53] to analyse trajectories.

### Isothermal titration calorimetry
Isothermal titration calorimetry (ITC) experiments were performed on a MicroCal PEAQ-ITC (Malvern). PvDBP-RII used in the ITC experiments was tagged N-terminally with a monomeric Fc domain (mono-Fc) and contained T257A, S353A and T422A substitutions to remove N-linked glycosylation sites; modifications are not in close proximity to the DARC binding site and hence should not interfere with binding. All synthetic peptides used in the ITC experiments were purchased from Cambridge Research Biochemicals (Billingham, UK) with the purity level of 95%. Prior to the measurement, mono-Fc tagged PvDBP-RII was extensively dialysed against 1x phosphate-buffered saline (PBS), and the peptides were dissolved in the same dialysis buffer to prevent buffer mismatch. The peptides were placed in the syringe at a concentration of ~130 μM for Ac-QLDFEDVWNSS-[Y(SO3H)]-GVNDSFPDG D-[Y(SO3H)]-DANLEL-NH$_2$, ~300 μM for Ac-QLDFEDVWNSSYGVNDSFP DGDYDANLEL-NH2, ~200 μM for Ac-QLDFEDVWNSS-[Y(SO3H)]-GVND SFPDGDYDANLEL-NH$_2$, and ~260 μM for Ac-QLDFEDVWNSSYGVNDS FPDGD-[Y(SO3H)]-DANLEL-NH$_2$, whereas the concentration of mono-Fc tagged PvDBP-RII in the cell was ~20 μM for all experiments. The peptide concentrations were determined by BCA assay using the Pierce Rapid Gold BCA Protein Assay Kit (ThermoFisher Scientific), and the protein concentration was determined by UV absorbance at 280 nm. The titrations were all performed with peptides in the syringe and mono-Fc tagged PvDBP-RII in the cell and consisted of a single 0.4 μl injection followed by 18 injections of 2 μl with injection duration of 4 s, injection spacing of 150 s, stir speed of 750 rpm, and reference power of 10 μcal/s. Experiments were conducted at 25°C in triplicate ($n = 3$), and the data are reported as the arithmetic mean ± SD. Fitting of the integrated titration peaks was performed with MicroCal PEAQ-ITC Analysis Software (Malvern) provided with the instrument, then all data were exported to and plotted in Prism 9 (Dotmatics).

### Small angle X-ray scattering
Mono-Fc tagged PvDBP-RII used for SAXS experiments was expressed in Expi293 cells (Thermo Scientific) as described above and purified using the CaptureSelect C-tagXL Affinity Matrix (Thermo Scientific). The elution was concentrated and buffer-exchanged into 20 mM HEPES, 150 mM NaCl pH 7.5 using a PD-10 Desalting Column (Cytiva) before TEV cleavage (1:100 TEV to protein ratio) overnight at 4 °C. After cleavage, mono-Fc was removed using Pierce Protein A agarose beads (Thermo Scientific). The flow-through containing PvDBP-RII was then mixed with DB1 or DB9 Fab (1.5:1 Fab to protein ratio) for 1 h before concentration and size-exclusion chromatography on a Superdex 200 Increase 10/300 GL column (Cytiva). The fractions containing PvDBP-RII with Fabs were concentrated or were mixed with DARC$_{19-47}$ Y30-S/Y41-S double sulfated peptides in case of the PvDBP-RII-DB9 Fab complex for 1 h prior to concentration.

Small-angle X-ray scattering (SAXS) data were collected at beamline B21 at the Diamond Light Source (Didcot, UK). Scattering was detected using X-rays at a wavelength of 0.99 Å and an Eiger 4 M detector with a detector-sample distance of 4.014 m. For the PvDBP-RII:DB9 Fab complexes with or without Y30-S/Y41-S double sulfated peptides, experiments were performed via two methods: (1) batch mode where 35 μl of samples at 6, 3, 1 mg/ml concentrations with corresponding matching buffers were injected directly into the capillary, using the EMBL Arinax sample handling robot, for data collection, and (2) SEC-SAXS where 50 μl of 6 mg/ml samples were gel filtered on the Shodex PROTEIN KW403-4F column (Shodex Denko Europe) equilibrated with 20 mM HEPES, 150 mM NaCl, pH 7.5 prior to data collection. The data for PvDBP-RII:DB1 complex were only collected using the SEC-SAXS method at the starting concentration of 6 mg/ml.

The data were processed using ScÅtter[54] with the ATSAS software suites[55]. For SEC-SAXS, buffer frames were averaged and subtracted from averaged frames of the complex peak fractions. The radius of gyration (Rg), distance distribution function P(r) and the maximum particle diameter (Dmax) were determined using ScÅtter. Twenty-nine ab initio initial bead models were generated with DAMMIF[55] from the ATSAS software suites in ScÅtter and averaged with DAMAVER[56] followed by refinement against the original data using DAMMIN[57]. The 20 Å envelope of the refined bead model was generated in Chimera[58]. Crystal structures of PvDBP-RII in complex with DB9 Fab[13] (PDB: 6R2S) overlayed with the Y41-S DARC peptide structure obtained in this study and the crystal structure of PvDBP-RII with DB1 Fab from the current study were fitted into the envelopes using Chimera. CRYSOL[55] was used to derive theoretical scattering data from monomeric and dimeric version of PvDBP-RII:DB9 or PvDBP-RII:DB1 complexes and to compare these data with the experimental results.

### Growth inhibition assays with transgenic *P. knowlesi*
To investigate DBP-DARC interactions in vitro, the variably sulfated DARC ectodomain peptides described above were used to competitively inhibit the host-parasite interaction between erythrocytic DARC and parasite-expressed DBP, which is essential for invasion and subsequent replication. Growth inhibition assays were run as previously described in refs. [13],[14] with parasite growth measured via lactate dehydrogenase (LDH) assay after one intraerythrocytic cycle (approx. 27 h) in the presence of various peptides, prepared in twofold dilution curves. Each 96-well plate included internal controls (infected, untreated erythrocytes and uninfected erythrocytes) which were used to calculate the percentage growth inhibition in each treatment well using the following formula (Eq. (2))[13,14]:

$$\% \, \mathrm{GIA} = 100 - \left( \left( \frac{\mathrm{OD_{sample} - OD_{uninfected\,RBCs}}}{\mathrm{OD_{infected\,RBCs} - OD_{uninfected\,RBCs}}} \right) \times 100 \right) \qquad (2)$$

Assays were conducted with a minimum of 2 technical replicates, and data presented represents averages from four separate experiments ($n = 2$ each using Fy$^a$ and Fy$^b$ donor red blood cells). Assays were performed using both *P. knowlesi* wild-type A1-H.1 (expressing PkDBPα), as well as transgenic *P. knowlesi* PvDBP$^{OR}$/PkDBPβγ$^{KO}$, which expresses the *P. vivax* DBP as an orthologue replacement of the homologous PkDBPα. IC$_{50}$ values were identified using a variable slope four-parameter logistic curve, calculated using GraphPad Prism v9.4.1.

### Reporting summary
Further information on research design is available in the Nature Portfolio Reporting Summary linked to this article.

## Data availability

Coordinates and structure factors generated in this study have been deposited in the Protein Data Bank with accession code 8A44. Source data are provided with this paper.

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

## Acknowledgements
M.K.H. is a Wellcome Trust Investigator (220797/Z/20/Z). R.M. and T.P. are funded by the Skaggs-Oxford graduate scholarship. B.G. is funded by the Wellcome PhD programme in Cellular Structural Biology. S.M.D. is supported by a UK Medical Research Council LID PhD studentship. R.W.M. was supported by the UK Medical Research Council (MRC Career Development Award MR/M021157/1). The authors thank Simon Draper for the coding sequence for antibody DB1 and the plasmid construct for monoFc-tagged PvDBP-RII. We thank the staff at Diamond beamline I04 for help with crystallographic data collection and staff at Diamond beamline B21 for help with SAXS data collection.

## Author contributions
R.M. produced proteins and R.M., E.L. and M.K.H. determined the crystal structure. T.P. conducted ITC and SAXS experiments. B.G. performed molecular dynamics analysis. S.M.D., F.M. and R.W.M. conducted para-site growth-inhibition experiments. R.W.M. and M.K.H. devised the study. M.K.H. drafted the manuscript and all authors discussed the results and commented on the manuscript.

## Competing interests
The authors declare no competing interests.
