## [Peer Review File · Nature Communications]

Structural basis for DARC binding in reticulocyte invasion by Plasmodium vivaxREVIEWER COMMENTS

Reviewer #1 (Remarks to the Author):

The manuscript by Moskovitz and Pholcharee et al describes the crystal structure of a sulphated peptide of DARC bound to the malaria parasite protein Plasmodium vivax Duffy Binding Protein (PvDBP). This sulphated DARC peptide is sulphated at Tyrosine 41 which has been shown to be involved in binding to PvDBP, so the crystal structure sheds light on the mechanism of ligand-receptor interaction. The authors also show that sulphation on tyrosine 30 and tyrosine 41 together on the DARC peptide provide the highest affinity binding to the parasite ligand which translates to higher inhibition of parasite invasion compared to no sulfation or single sulfation. In addition, the authors describe a crystal structure of DB1 an inhibitory antibody bound to PvDBP.

We ask for some clarification on the structural biology approaches and some easier representation of the interactions.

Comments:

1. The authors state in Line 40 that “The parasite receptor for DARC is the Duffy binding protein...” whereas they describe DARC as the human receptor for PvDBP in Line 5. We suggest that the term ‘ligand’ would be more intuitive for the parasite interaction partner and to change Line 40 to read “The parasite ligand for DARC...”.

2. In Line 59-60, the authors state: ‘The only structural insight into this interaction comes from structures of PvDBP-R11 bound to an 11-residue helical peptide (DARC19-30)’.

a. Please provide the missing reference for this statement.

b. However, if the authors are referring to Batchelor et al. PLOS Pathogens 2014 the statement is misleading. In the structures published by Batchelor the DARC peptides used for crystallisation contained residues 16-43 or 14-43 respectively (which are around 28 to 30 residues in length), but that the electron density was observed for eleven amino acids, namely residues 19-30. Please clarify as it seems important for the contribution of this study to the field if the previously structures used smaller peptides or peptides of similar length to discuss the contribution of the sulphated tyrosine(s) for binding to PvDBP.

3. Figure 1, panel c and d:

- a. Please specify/ indicate which electron density map (2F_c-F_o? Omit map?) and sigma level was used for DARC.
- b. For electrostatic surface potential: What kT values were chosen to colour the surface?
- c. Can the authors please use the colour scheme in Figure 2b,d to highlight the Sulphate of Y41 so it is easier to view in the figure?

4. In Line 84-86: "This was purified and assessed by mass spectrometry, with the predominant peak having the mass expected for the peptide with the addition of two sulphates."

Please provide the mass spectrometry results of purified peptide as a supplementary figure.

5. Line 94: Which of the two structures in reference 29 (Batchelor et al, PloS Pathogen 2014) were used for the alignment?

- a. Please provide the relevant PDB ID.
- b. Please provide an indication of how well do the structures align in general using rmsd values.

6. In Line 101-102: Can the authors describe if D42 is involved in any additional interactions with PvDBP?

7. Line 109: "the Y30 side chain was clearly visible in the electron density, but no evidence was seen for sulphation..."

- a. This appears to be a vague statement and can the authors please elaborate? Sulphate is relatively electron rich. What is the effect on R-factors/ electron density when the authors use a sulphated Y30 in the refinement process? What is the B-factor of the Y30 side chain? As the sulphate is covalently linked to the side chain it would be surprising if the aromatic part of the side chain has low B-factors and the sulphate very high B-factors so that it is completely absent and not visible in the density.
- b. To our knowledge tyrosine O-sulphation is stable / irreversible in vivo. As the double-sulphated peptide has a higher affinity compared to Y41-S and as the mass spectrometry result of the purified DARC peptide used for crystallisation (line 84-86) suggested that the majority of it has two sulphates we find it surprising that the authors did not follow up (experimentally) on their finding that the bound peptide shows only Y41 sulphation. For example, mass spectrometry of PvDBP-DARC crystals could provide more insights here.

8. Line 111-113: 'The side chain of Y30 is involved in crystal contacts'. However, there is no elaboration on the crystal contacts. Can the authors please provide more information on what kind of interaction

and what part of the side chain is involved? Does this allow any conclusion about the sulphation state of this residue? Supplementary figure would help here.

9. Line 116: "...where sulphate makes direct salt bridges with K301 and R304 (Figure 1c)".

a. Please label these interactions in Figure 1c.

b. Some of the surrounding residues are shown in Fig 2b but the shown residues are not labelled which makes it impossible to follow the text. Please label the relevant residues to allow easier visualisation of the interaction.

10. Figure 2b (right): Simulations of non-sulphated forms of Y41 are shown on the right. Why is there a sulphate shown in the figure?

11. Figure 3: Please add error values for ITC measurements. The KD values of individual measurements vary quite a lot of Supplemental Fig 1. Can the authors provide any comment on the wide range of measurements?

12. Line 138: "We next assessed the number of interactions formed, and the fraction of the simulation during which each interaction occurs."

a. Can the authors please describe what kind of contacts did the authors look at? H-bonds? Salt bridges? Van-der Waals?

b. Can they please discuss why they chose a 4.5 angstrom cut-off?

13. In Line 145, the authors state: "Y41 makes a strong interaction with R305."

a. Is it R305 or R304 in Figure 2f?

b. Can the authors please label the relevant Arginine (304 or 305) in Figure 2b?

c. How strong is this interaction? Can the authors describe which calculation was used to classify this interaction to be strong?

14. Figure 4B. How was dimer model generated? If a published crystal structure was used, please state it here.

15. Supp Table 1: Please provide B factors for individual chains in addition to average B-factor or instead.

16. Minor: Inconsistent usage of dimerization or dimerisation; spaces between many values and their units missing, and ml vs mL.

Reviewer #2 (Remarks to the Author):

The manuscript by Moskovitz et al. is interesting and makes important contributions to the design of vaccines and therapeutics for *Plasmodium vivax* infection. The article is well-written and mostly clarified some fog in the field. However, a few important scholarly and technical concerns must be addressed. I would like to see the revised paper as critical areas of concern must be addressed to enhance value of this paper:

1. Previous studies reported the sulphation of tyrosine residues in DARC, and it was well known that sulphation of DARC enhances binding to DBP – indeed the first paper on this had identified the site which now has been verified (Singh et al. 2006, Nature). The paper should thus add how their new experimental evidence is in sync with the proposed binding site and cite that study.
2. The paper by Yogavel et al. 2018 (Front Mol Biosci. 2018, 5:78) that suggested two distinct DARC binding sites in Pk/Pv-DBLs based on site-directed mutagenesis data. It was also a modelling study using sulphate and phosphate-bound crystal structures of Pk/Pv-DBLs that agrees with the authors current data. Hence it should be cited as it had cogently dissected the receptor-ligand interactions that seem to be validated in the Moskovitz et al work.
3. The present report (co-crystal structure and SAXS experiments) do not find dimerization of the PvDBP-RII-DARC-antibody complex as was proposed by Tolia et al. Additionally, Tolia et al had proposed that DARC peptide binding itself drives dimerization. However, whether that is the case or not needs to be addressed by the current authors by simply testing via GPC or SAXS the monomeric/dimeric state of DBL when bound to the DARC peptide.
4. Data on DARC polymorphic (D42G) purification, binding and inhibition assays and MD calculation for PvDBP-DARC (D42G) based on the present DARC peptide structure can be added or addressed.
5. The author stated that the mass spectrometry analysis data for the DARC peptide had a dominant peak for the expected peptide (60 residues) with two sulphates. However, in the crystal structure, sulphonation was observed only on residue Y41. Why not for Y30?

6. An appropriate word should be used sulphation or sulphonation

7. L21: 75% of malaria in the Americas and 50% in South East Asia.

The total percentage of malaria in the Americas and Southeast Asia should be written correctly.

8. L23: Recent vaccine trials in Africa – anyway these are not for vivax so what's the relevance?

9. L91: 2.49Å resolution to be written as 2.49 Å resolution.

A blank space should be between the number and the unit. This should be followed throughout the manuscript and in the figure legends.

10. Few other places: L157: 58nM, L153: 23µM, L160: 552nM, L138: 4.5Å, L492: 0.72µM, L493: 0.79µM, 2.99µM, L494: 3.92µM

11. L85: Need to add supporting data for mass spectrometry.

12. L116 and L145: There is a mismatch for the Arg residue label (R304/R305). R304 is correct

13. L123: There is no comparison between sulphated (Y30 and Y41) and un-sulphated models. The author should attempt to compare the free energy surfaces and several interactions for sulphated and un-sulphated models, which may provide additional information.

14. In Figure 1c), the sulphated Y41, i.e. Y41-S, should be marked. The contour level and type of electron density map should be included in the Figure 1 legend.

15. There are four discrepancies between the depository and actual sequence for DARC. No information about this is in the manuscript. Figure 4: domain diagram showing residues crystallized could be added if feasible.

Reviewer #3 (Remarks to the Author):

The present work proposes a clear idea on the full interaction between the ectodomain of DARC and the PvDBP-RII for their critical interaction for red blood cell invasion in Plasmodium vivax

Further, the methodology is clear and pose reproducible datasets

Key finding are proven in MD and has been extended to experiments- which is quite appreciable

Significant findings - Sulphated Y41 and Y30 interaction, stability and ambient explanation was given with supportive evidence.

One question would be - Does the structure attained global minima and it can be depicted as FES landscape in the supplementary information. State of the bound ligand during the course of the simulation can be high-lightened to understand the dynamic of the complex.

Appreciate the authors for their clear methodology, result and discussion.

Reviewer #1 (Remarks to the Author):

The manuscript by Moskovitz and Pholcharee et al describes the crystal structure of a sulphated peptide of DARC bound to the malaria parasite protein Plasmodium vivax Duffy Binding Protein (PvDBP). This sulphated DARC peptide is sulphated at Tyrosine 41 which has been shown to be involved in binding to PvDBP, so the crystal structure sheds light on the mechanism of ligand-receptor interaction. The authors also show that sulphation on tyrosine 30 and tyrosine 41 together on the DARC peptide provide the highest affinity binding to the parasite ligand which translates to higher inhibition of parasite invasion compared to no sulfation or single sulfation. In addition, the authors describe a crystal structure of DB1 an inhibitory antibody bound to PvDBP.

We ask for some clarification on the structural biology approaches and some easier representation of the interactions.

Comments:

1. The authors state in Line 40 that “The parasite receptor for DARC is the Duffy binding protein...” whereas they describe DARC as the human receptor for PvDBP in Line 5. We suggest that the term ‘ligand’ would be more intuitive for the parasite interaction partner and to change Line 40 to read “The parasite ligand for DARC...”.

We appreciated the reviewer’s point, as calling both DARC and PvDBP a ‘receptor’ is confusing. We prefer not to use ligand for PvDBP, as to our mind the ligands for DARC are chemokines etc. We have therefore changed ‘receptor’ to ‘binding partner’ in line 40, where we refer to PvDBP.

2. In Line 59-60, the authors state: ‘The only structural insight into this interaction comes from structures of PvDBP-R11 bound to an 11-residue helical peptide (DARC19-30)’.

a. Please provide the missing reference for this statement.

b. However, if the authors are referring to Batchelor et al. PLOS Pathogens 2014 the statement is misleading. In the structures published by Batchelor the DARC peptides used for crystallisation contained residues 16-43 or 14-43 respectively (which are around 28 to 30 residues in length), but that the electron density was observed for eleven amino acids, namely residues 19-30. Please clarify as it seems important for the contribution of this study to the field if the previously structures used smaller peptides or peptides of similar length to discuss the contribution of the sulphated tyrosine(s) for binding to PvDBP.

The reviewer is correct. This statement did refer to Batchelor et al 2014 and the construct used was longer than 11 residues (residues 14-43). We have expanded this sentence (now lines 60-62) to make this point more completely and to add the missing reference.

3. Figure 1, panel c and d:

a. Please specify/ indicate which electron density map (2Fc-Fo? Omit map?) and sigma level was used for DARC.

This information has now been added to the figure legend.

b. For electrostatic surface potential: What kT values were chosen to colour the surface?

We have clarified in the figure legend that pymol was used for estimation of the electrostatic surface potential and have added the scale from pymol into the figure. However, we would caution readers against using these estimates. They are intended as illustrative of charge distribution, rather than to provide precise values.

c. Can the authors please use the colour scheme in Figure 2b,d to highlight the Sulphate of Y41 so it is easier to view in the figure?

We have coloured the atoms in Figure 1c and d to match those in Figure 2.

4. In Line 84-86: “This was purified and assessed by mass spectrometry, with the predominant peak having the mass expected for the peptide with the addition of two sulphates.”

Please provide the mass spectrometry results of purified peptide as a supplementary figure.

We have added the mass spectrum as Supplementary Figure 1. The mass of 10072 is consistent with the peptide mass of 9918, together with two sulphates and the expected deprotonation of this acidic peptide.

5. Line 94: Which of the two structures in reference 29 (Batchelor et al, PloS Pathogen 2014) were used for the alignment?

a. Please provide the relevant PDB ID.

We have added the PDB code used into the legend of Figure 1b (line 493).

b. Please provide an indication of how well do the structures align in general using rmsd values.

We have added the RMSD for the helical residues in lines 106-107.

6. In Line 101-102: Can the authors describe if D42 is involved in any additional interactions with PvDBP?

We have now stated in line 114 that there is no clear interaction of D42 with PvDBP-R11. This is also shown in our new interaction table – supplementary table 2. As mentioned in our response to reviewer 2, our structure does not rationalise the effect of the D42G polymorphism.

7. Line 109: “the Y30 side chain was clearly visible in the electron density, but no evidence was seen for sulphation...”

a. This appears to be a vague statement and can the authors please elaborate? Sulphate is relatively electron rich. What is the effect on R-factors/ electron density when the authors use a sulphated Y30 in the refinement process? What is the B-factor of the Y30 side chain? As the sulphate is covalently linked to the side chain it would be surprising if the aromatic part of the side chain has low B-factors and the sulphate very high B-factors so that it is completely absent and not visible in the density.

b. To our knowledge tyrosine O-sulphation is stable / irreversible in vivo. As the double-sulphated peptide has a higher affinity compared to Y41-S and as the mass spectrometry result of the purified DARC peptide used for crystallisation (line 84-86) suggested that the majority of it has two sulphates we find it surprising that the authors did not follow up (experimentally) on their finding that the bound peptide shows only Y41 sulphation. For example, mass spectrometry of PvDBP-DARC crystals could provide more insights here.

We agree that it is surprising that we do not observe clear electron density on Y30 for a sulphate modification (as shown in Figure 1c) particularly when the mass for the peptide (now Supplementary Figure 1) clearly indicates two sulphate modifications. This is not due to disorder of Y30 which has a B factor of 63.2\AA^2 , compared with 87.2\AA^2 for Y41S. This lack of sulphate density, together with Y30 forming crystal contacts rather than interacting with PvDBP, were some of the reasons why we ran a comprehensive molecular dynamics simulation experiment. While the crystal structure does not give a final view of the role of the sulphate on Y30, or show contact between Y30 and PvDBP-R11, the combination of molecular dynamics simulations with affinity measurements and growth inhibitory activity measurements combine to give a consistent picture, showing that Y30 does bind to PvDBP-R11 in solution, but that the sulphate on Y30 has a smaller effect on affinity than that on Y41 and does not have a strongly preferred binding site to PvDBP-R11. We have added a sentence in lines 151-154 to make the point that the molecular dynamic simulation provides insight which is lacking due to the crystal packing around Y30.

8. Line 111-113: ‘The side chain of Y30 is involved in crystal contacts’. However, there is no elaboration on the crystal contacts. Can the authors please provide more information on what kind of interaction and what part of the side chain is involved? Does this allow any conclusion about the sulphation state of this residue? Supplementary figure would help here.

As this is not a biologically relevant interaction, it is our preference that we should not describe this crystal contact in any further detail in the manuscript. Readers will be able to look at this for themselves using the deposited PDB should they wish. On the sulphation point, see above.

9. Line 116: “...where sulphate makes direct salt bridges with K301 and R304 (Figure 1c)”.

a. Please label these interactions in Figure 1c.

b. Some of the surrounding residues are shown in Fig 2b but the shown residues are not labelled which makes it impossible to follow the text. Please label the relevant residues to allow easier visualisation of the interaction.

We have now labelled these residues in Figure 2. We prefer not to additionally label them in Figure 1, which would involve changing the figure structure and, in our view, losing clarity.

10. Figure 2b (right): Simulations of non-sulphated forms of Y41 are shown on the right. Why is there a sulphate shown in the figure?

The author is correct to note that this was confusing. The crystal structure had been shown in light pink as a reference and it is from this structure that the sulphate derives. We had omitted this fact from our figure legend. This is now added to the legend for 2b and 2d.

11. Figure 3: Please add error values for ITC measurements. The KD values of individual measurements vary quite a lot of Supplemental Fig 1. Can the authors provide any comment on the wide range of measurements?

The errors for each of the individual ITC measurements are shown in the legend for Supplementary Figure 2. We agree that the measurements vary more than we would normally expect. This is most likely because they are true biological replicates, conducted with batches of peptide and protein made up separately. Despite this variation, the conclusions about the relative effects of sulphation hold.

12. Line 138: “We next assessed the number of interactions formed, and the fraction of the simulation during which each interaction occurs.”

a. Can the authors please describe what kind of contacts did the authors look at? H-bonds? Salt bridges? Van-

der Waals?

b. Can they please discuss why they chose a 4.5 angstrom cut-off?

The procedure which was used to analyse these molecular dynamics simulations does not distinguish between different types of interactions but instead is a measure of contact, as indicated purely by distance. We have clarified this through changes to lines 155-156, in which we have changed 'interaction' to 'contact'.

The selection of 4.5Å as a heavy atom distance as the cut-off is the standard in the field for defining contacts as it allows all of the standard interactions found within proteins to be included (i.e. Méndez R et al (2023) PMID: 12784368; Abdel-Azeim S et al (2014) PMID: 25077693 and Leem J et al (2022) PMID: 35845836).

As this data does not allow us to specify specific interactions, we have instead included a new interactions table in Supplementary Table 2, in which we list the interactions found in the crystal structure.

13. In Line 145, the authors state: "Y41 makes a strong interaction with R305."

a. Is it R305 or R304 in Figure 2f?

This was incorrect and it has been corrected, now line 163, to R304.

b. Can the authors please label the relevant Arginine (304 or 305) in Figure 2b?

This is now labelled.

c. How strong is this interaction? Can the authors describe which calculation was used to classify this interaction to be strong?

The reviewer is correct to challenge the word 'strong' as we did not measure the strength of this individual interaction. We have instead changed this in lines 163 to clarify that it is persistence rather than strength which we assess, as the interaction is retained throughout the simulation.

14. Figure 4B. How was dimer model generated? If a published crystal structure was used, please state it here.

This has been added to the legend for figure 4 (line 562).

15. Supp Table 1: Please provide B factors for individual chains in addition to average B-factor or instead.

We have added these, as requested.

16. Minor: Inconsistent usage of dimerization or dimerisation; spaces between many values and their units missing, and ml vs mL.

We have corrected these, where we spotted them, and are confident that they will be adjusted to whatever the journal house style is during the type-setting process.

Reviewer #2 (Remarks to the Author):

The manuscript by Moskovitz et al. is interesting and makes important contributions to the design of vaccines and therapeutics for *Plasmodium vivax* infection. The article is well-written and mostly clarified some fog in the field. However, a few important scholarly and technical concerns must be addressed. I would like to see the revised paper as critical areas of concern must be addressed to enhance value of this paper:

We thank for the reviewer for these positive comments and have made the suggested changes.

1. Previous studies reported the sulphation of tyrosine residues in DARC, and it was well known that sulphation of DARC enhances binding to DBP – indeed the first paper on this had identified the site which now has been verified (Singh et al. 2006, Nature). The paper should thus add how their new experimental evidence is in sync with the proposed binding site and cite that study.

2. The paper by Yogavel et al. 2018 (Front Mol Biosci. 2018, 5:78) that suggested two distinct DARC binding sites in Pk/Pv-DBLs based on site-directed mutagenesis data. It was also a modelling study using sulphate and phosphate-bound crystal structures of Pk/Pv-DBLs that agrees with the authors current data. Hence it should be cited as it had cogently dissected the receptor-ligand interactions that seem to be validated in the Moskovitz et al work.

The reviewer is correct and we are very happy to cite both of these papers about the closely related *Plasmodium knowlesi* homologue of PvDBP-R11, as they are excellent studies which are extremely relevant to this work. We have now described key conclusions from both Singh and Yogavel in the introduction (lines 58-59 and 66-70) and have indicated in line 112 that the proposed sulphation binding site in these studies is the same as the site we observe.

3. The present report (co-crystal structure and SASX experiments) do not find dimerization of the PvDBP-R11-

DARC-antibody complex as was proposed by Tolia et al. Additionally, Tolia et al had proposed that DARC peptide binding itself drives dimerization. However, whether that is the case or not needs to be addressed by the current authors by simply testing via GPC or SAXS the monomeric/dimeric state of DBL when bound to the DARC peptide.

The experiment presented in Supplementary Figure 4b is essentially what the reviewer proposes, although this was not fully clear from the figure legend. This experiment was conducted in the presence of the double sulphated DARC peptide which matched that used in the ITC experiments in figure 3. We have clarified this in lines 225, 226-7 and 869. Admittedly, this experiment has been conducted in the presence of the fab fragment of the DB9 antibody, while the study by Tolia et al was performed without the antibody. In our hands, as stated in lines 224-226, this complex, in the absence of DB9, was sticky and showed substantial aggregation in SAXS experiments. The data from Tolia et al does show signs of aggregation, which could possibly explain the difference. We are not concerned about having DB9 present as it binds at the opposite end of PvDBP from the proposed dimerisation interface. Therefore the data shown in Supplementary Figure 3b is as close as we can technically get to the experiment proposed by the reviewer without the confounding effect of aggregation.

4. Data on DARC polymorphic (D42G) purification, binding and inhibition assays and MD calculation for PvDBP-DARC (D42G) based on the present DARC peptide structure can be added or addressed.

In this study, we do not focus on the effect of the D42G polymorphism as we do not have conclusive data which demonstrates how this polymorphism alters reticulocyte invasion by *P. vivax*. We did produce the double-sulphated D42G peptide and assessed its binding to PvDBP-R11 by ITC. However, we did not see any effect of D42G on affinity (see below). Indeed, no clear interactions are observed between residue 42 and PvDBP-R11 in the crystal structure, now stated in line 114 and supplementary table 2. Experiments to understand this are underway. However, these are complex, as residue 42 might be exerting its effect indirectly through changes in DARC expression and/or distribution and/or modification, and so this study will not be completed in a time frame to include in this manuscript. We therefore prefer to not include the D42G ITC data in this manuscript, where it will not contribute to firm conclusions, but to wait to publish this data as part of a conclusive study in which we answer this question.

$$K_D 86.7 \pm 45.2 \text{ nM}$$

5. The author stated that the mass spectrometry analysis data for the DARC peptide had a dominant peak for the expected peptide (60 residues) with two sulphates. However, in the crystal structure, sulphonation was observed only on residue Y41. Why not for Y30?

Please see the response to reviewer 1 point 7.

6. An appropriate word should be used sulphation or sulphonation.

Our preference is 'sulphonation' which we believe is used throughout.

7. L21: 75% of malaria in the Americas and 50% in South East Asia.

The total percentage of malaria in the Americas and Southeast Asia should be written correctly.

We have checked the reference given here and have provided more precise values in lines 20-21.

8. L23: Recent vaccine trials in Africa – anyway these are not for vivax so what's the relevance?

Apologies. We did not understand the point made here by the reviewer. Line 23 just says that it would be good to have a vaccine. Which it would.

9. L91: 2.49Å resolution to be written as 2.49 Å resolution.

A blank space should be between the number and the unit. This should be followed throughout the manuscript

and in the figure legends.

10. Few other places: L157: 58nM, L153: 23 μ M, L160: 552nM, L138: 4.5 \AA , L492: 0.72 μ M, L493: 0.79 μ M, 2.99 μ M, L494: 3.92 μ M

We have corrected these, where we spotted them, and are confident that they will be adjusted to whatever the journal house style is during the type-setting process.

11. L85: Need to add supporting data for mass spectrometry.

We have added the mass spectrum as Supplementary Figure 1. The mass of 10072 is consistent with the peptide mass of 9918, together with two sulphates and the expected deprotonation of the acidic peptide.

12. L116 and L145: There is a mismatch for the Arg residue label (R304/R305). R304 is correct.

This is corrected, now on line 163.

13. L123: There is no comparison between sulphated (Y30 and Y41) and un-sulphated models. The author should attempt to compare the free energy surfaces and several interactions for sulphated and un-sulphated models, which may provide additional information.

These simulations aimed to assess the free energy surfaces for individual residues (Y30 and Y41) in the presence and absence of sulphation, rather than for the complete DARC peptide (see lines 134-136 and 342-357). This was measured in the context of the binding mode observed for the overall DARC peptide in the crystal structure. The three simulations conducted here, Y30S, Y41S and double sulphated, allowed us to determine this information without simulating the unsulphated peptide. Indeed, to obtain this information for the peptide which lacked Y41 sulphation, it was necessary to provide a restraint on the backbone of the DARC peptide to prevent dissociation, as is consistent with the substantial contribution of Y41S to the binding affinity. In a simulation of unsulphated peptide, we would also need to restrain the backbone, as this peptide has an even weaker affinity. This backbone restraint would mean that the sulphation state of Y41 and Y30 will not affect the dynamics of the other tyrosine and a simulation for the unsulphated peptide will not provide additional information. We have now emphasised that this is a side-chain specific free energy surface in lines 135.

14. In Figure 1c), the sulphated Y41, i.e. Y41-S, should be marked. The contour level and type of electron density map should be included in the Figure 1 legend.

These changes have been made

15. There are four discrepancies between the depository and actual sequence for DARC. No information about this is in the manuscript. Figure 4: domain diagram showing residues crystallized could be added if feasible.

These discrepancies have been resolved. Residues 4, 51, and 54, which lie outside the binding site, were changed from C to A to prevent disulphide formation which prevented crystallisation and the error for 47 has been corrected. An updated validation report is included.

Reviewer #3 (Remarks to the Author):

The present work proposes a clear idea on the full interaction between the ectodomain of DARC and the PvDBP-R11 for their critical interaction for red blood cell invasion in Plasmodium vivax.

Further, the methodology is clear and pose reproducible datasets.

Key finding are proven in MD and has been extended to experiments- which is quite appreciable.

Significant findings - Sulphated Y41 and Y30 interaction, stability and ambient explanation was given with supportive evidence.

One question would be - Does the structure attained global minima and it can be depicted as FES landscape in the supplementary information. State of the bound ligand during the course of the simulation can be highlighted to understand the dynamic of the complex.

We thank the reviewer for this suggestion. We interpret this to mean a request to compare the observed χ_1 angles in the crystal structure with the free energy surface from the MD simulations. We have therefore measured the χ_1 angles for Y30 and Y41 in the crystal structure and have added these two the text (lines 144 and 149-151) as well as pointing them out in Figure 2a and c. This emphasises that both residues in the crystal structure lie in locations found in local energy minima in the molecular dynamics simulations.

Appreciate the authors for their clear methodology, result and discussion.

We thank for the reviewers for their many positive comments.